# Analysis of Quantitative Light-Induced Fluorescence Images for the Assessment of Bacterial Activity and Distribution of Tongue Coating

**DOI:** 10.3390/healthcare11020217

**Published:** 2023-01-11

**Authors:** Yu-Rin Kim, Hyun-Kyung Kang

**Affiliations:** Department of Dental Hygiene, Silla University, 140 Baegyang-daero 700 Beon-gil, Sasang-gu, Busan 46958, Republic of Korea

**Keywords:** bacteria, halitosis, oral health, tongue coating, quantitative light-induced fluorescence

## Abstract

This study aimed to verify and to validate the correlation between, and validate the assessment of, bacterial activity and the distribution of tongue coating, by analyzing quantitative light-induced fluorescence (QLF) images for the diagnosis of oral malodor. Oral malodor was measured using the Twin Breasor II, and bacterial activity and the distribution of tongue coating were investigated using QLF images. Bacterial activity and the distribution of tongue coating were quantitatively analyzed by dividing the tongue into six areas using the TB01 1.05 software. ROC curves were generated using MedCalc^®^ software to validate the diagnosis of oral malodor, the testing of bacterial activity, and the distribution of tongue coating. Bacterial activity and the distribution of tongue coating showed a very strong association with each other (r = 0.937, *p* < 0.01), and were significantly higher in the oral malodor group (*p* < 0.05). The distribution of tongue coating was highly correlated with H_2_S (r = 0.223, *p* < 0.01) but not associated with CH_3_SH. Sensitivity, specificity, and the area under the curve confirmed the high accuracy of this method in assessing bacterial activity and the distribution of tongue coating in the diagnosis of oral malodor. Since QLF images provide significant accuracy during quantitative analysis in the identification of bacterial activity and the distribution of tongue coating, systematic management of tongue coating and reduced oral malodor can be achieved by actively using QLF images and oral malodor measurement.

## 1. Introduction

Since social communication has become essential in establishing diverse interpersonal relations, the management of oral malodor is highly important for social and health reasons [1]. Oral malodor is an unpleasant smell from the oral cavity due to physiological or various pathological causes [2]. As a result of analyzing objective results for halitosis, prevalence rates of 10% to 20% of the population in the United States [3] and 20% to 34% in China have been reported [4]. Oral malodor can be classified into genuine halitosis, pseudo-halitosis, and halitophobia. Genuine halitosis refers to oral malodor that is recognized by others. Pseudo-halitosis is defined as a self-perceived complaint of oral malodor that is not recognized by others [5,6] and shows no symptoms. Halitophobia refers to a patient who believes that he or she has oral malodor even after the treatment of genuine or pseudo-halitosis [7].

The causes of oral malodor can be divided into the following two major categories: intraoral and extraoral. Intraoral causes include dental caries, periodontal diseases, poor dental restorations, and excessive bacterial colonies on the surface of the tongue. Extraoral causes include nasal and respiratory diseases, gastrointestinal diseases, systemic diseases, and medication [8]. Among these causes, approximately 90% of all halitosis cases originate intraorally [9].

Oral malodor originates from the decomposition of peptides and protein-derived amino acids included in gingival crevicular fluid, blood, detached epithelial cells, saliva, and food debris by intraoral anaerobic Gram-negative bacteria [10,11]. Volatile sulfur compounds (VSCs) are generated from this process and exhaled, leading to halitosis [8]. Rosenberg [12] reported that hydrogen sulfide (H_2_S), methyl mercaptan (CH_3_SH), and dimethyl sulfide account for 90% of all VSCs as the main components and suggested that halitosis becomes more severe as the concentration of VSCs increases. The clinical diagnostic criteria for oral malodor are based on the concentration of VSCs that cause unpleasantness to others as follows: H_2_S ≥ 1.5 ng/10 mL and CH_3_SH ≥ 0.5 ng/10 mL [13].

The common intraoral sites that induce oral malodor are the tongue and the subgingival area. In particular, about 60% of all halitosis cases originate from the coating on the tongue surface [14,15]. The tongue coating is formed on the tongue, which has a broad surface due to papillary structures. This occurs due to the accumulation of bacteria, a large number of epithelial cells detached from the oral cavity, blood metabolites, and food debris. Oho et al. [16] reported that the level of tongue coating was higher in patients with oral malodor than in the control group, which verifies the association between the presence of tongue coating and oral malodor. Moreover, microorganisms, tongue coating, and gingival crevicular fluid were reported to be the main factors that increase halitosis in patients with periodontal diseases [17]. In addition, it was stated that the mucosal surface of the tongue is the most common site where oral malodor is produced [18]. However, since the thickness and color of the tongue coating as measured by certain medical methods demonstrate a normal thickness in patients with oral malodor [19], it is currently difficult to measure halitosis by using only the amount of tongue coating. In addition, even though the method proposed by Winkel et al. [20] can be used to measure oral malodor, it has not been used widely owing to problems in quantification and standardization, as the method is based on the experience and knowledge of the examiners, which makes it difficult to produce objective and reproducible results. Hence, it is necessary to confirm the association of tongue coating with oral malodor by investigating not only its distribution but also the activity of halitosis-inducing bacteria. Therefore, diagnostic techniques to quantitatively identify tongue coating and bacterial activity are required.

The Qraycam Pro (AIOBIO, Seoul, Republic of Korea) shows strong fluorescence for porphyrin compounds produced by oral microorganisms and tends to show stronger red fluorescence (RF) as halitosis-inducing Gram-negative bacteria increase [21]. Porphyrin is produced by the process of hemoglobin degradation by bacteria [22,23], and intraoral microorganisms can fluoresce through various processes, including porphyrin metabolism. In this study, the Qraycam Pro was used as the diagnostic instrument to quantify the distribution of tongue coating and bacterial activity.

This study aimed to verify and to validate the correlation between, and validate the assessment of, bacterial activity and the distribution of tongue coating by analyzing Qraycam Pro images for the diagnosis of oral malodor.

## 2. Materials and Methods

### 2.1. Participants

Ninety-five patients who visited the Seongso dental clinic located in Busan between October 2019 and January 2020 were informed about the aim of the study. Patients who volunteered to participate in the study were asked to complete a self-administered questionnaire. To determine the adequacy of the sample size, the minimal sample size for the correlation analysis was calculated using a two-tailed test with a power of 1−β = 0.8 and an effect size of 0.3 (medium) using G*Power 3.1 software. The required sample size was found to be 84. Assuming a dropout rate of 10%, a total of 92 patients were included. Of the 92 subjects, the final 89 subjects, excluding 3 subjects with errors in the measured results, were selected for analysis. This study was approved by the Institutional Review Board at Silla University (No. 1041449-201912-HR-004, 13 December 2019).

### 2.2. Questionnaire

In the self-administered questionnaire, subjects were asked about their characteristics, including their gender, age, type of cohabitation, and the medications they used. In addition, drinking habits, smoking habits, exercise habits, and recent health status were investigated. Using a five-point scale, subjects were also asked to rate their own drinking, smoking and exercise habits, and their current health status. A lower score indicated more drinking and smoking along with less exercise and lower current health status.

### 2.3. Test for Oral Malodor

We measured H_2_S and CH_3_SH using the Twin Breasor II (iSenLab Inc., Seoul, Republic of Korea) to determine the level of oral malodor. Participants held a dedicated straw in their mouths and inhaled through the mouth for 10 s after nasal respiration for 50 s. The analysis was performed for a duration of 150 s after the inhalation, and the results are represented in the form of graphs and values. According to the results, when the value of the two gases (H_2_S and CH_3_SH) representing oral malodor was 0, no oral malodor was determined. In addition, when H_2_S was 1.5 or more and CH_3_SH was 0.5 or more, it was determined that there was oral malodor (Figure 1).

### 2.4. Assessment of Bacterial Activity and the Distribution of Tongue Coating on the Dorsal Surface

Qraycam Pro (AIOBIO, Seoul, Republic of Korea) is a device that uses the QLF technique to detect red fluorescence from porphyrin metabolites secreted by intraoral bacteria and to investigate bacterial activity and tongue coating [24]. The older the dental plaque, the more red fluorescence it shows [4]; this can provide significant visual assistance when educating patients (Figure 2).

To determine the bacterial activity and quantitative distribution of tongue coating, the tongue of each participant was positioned as extra-orally as possible, and Qraycam Pro images were obtained. R represents the average value of red fluorescence in the analysis area (in the yellow border area at the top left of the attached picture) (Figure 3). Coverage is shown in the analysis result provided in the contents of Figure 3, and it is expressed as the percentage of red fluorescence area compared with the entire analysis area. The tongue images were divided into six areas (A, B, C, D, E, F) using the TB01 1.05 analyzer program, and the red fluorescence value of each area and the distribution of tongue coating and bacterial activity were analyzed quantitatively.

### 2.5. Data Analysis

The data were analyzed using IBM SPSS ver. 26.0 (IBM Co., Armonk, NY, USA) with an α of 0.05. The participants were categorized into a non-oral malodor group (VSCs = 0) and an oral malodor group (H_2_S ≥ 1.5 and CH_3_SH ≥ 0.5) according to the results of the oral malodor test. For each of the two groups, the characteristics of the subjects (gender, age, type of cohabitation, the medications they used, drinking, smoking, exercise habits, and their current health status) were assessed with a chi-squared test or an independent *t*-test. Independent *t*-tests were used to compare the mean amount of tongue coating in each of the 6 areas of the tongue, bacterial activity, the distribution of tongue coating throughout the tongue, and VSCs in each of the two groups. Additionally, a correlation analysis was conducted to determine the relationship between oral malodor, bacterial activity, and tongue coating distribution in all subjects. Receiver operating characteristic (ROC) curves were calculated using MedCalc^®^ software (ver. 8.1.1.0, MedCalc Software, Ostend, Belgium) to validate the diagnosis of oral malodor, the identification of bacterial activity, and the distribution of tongue coating. The sensitivity and specificity of the method used to analyze bacterial activity and the distribution of tongue coating in the diagnosis of oral malodor were calculated, and the optimal cut-off value for the diagnosis of oral malodor was determined according to the area under the curve (AUC) based on the highest sum of sensitivity and specificity for each threshold.

## 3. Results

### 3.1. Comparison of Characteristics of the Non-Oral Malodor Group and Oral Malodor Group

The participants with VSCs of 0 were classified as the non-oral malodor group, and those with H_2_S ≥ 1.5 and CH_3_SH ≥ 0.5 were classified as the oral malodor group. As a result of confirming the demographic characteristics of the two groups, there was no significant difference in all variables, which ensured the homogeneity of the two groups. There were more females than males in both the non-oral malodor group and the oral malodor group, and there were more participants aged 65 years or older than participants who were younger than 65 years old. In relation to the cohabitation arrangements of the participants, the categories “alone” and “couple without children” accounted for the majority of the non-oral malodor and oral malodor groups, respectively. For medication being taken, “no” was the most frequent response in both groups. The oral malodor group drank more alcohol and smoked more than the non-oral malodor group. The oral malodor group exercised more than the non-oral malodor group, but their current state of health was no better (Table 1).

### 3.2. Comparison of Distribution of Tongue Coating by Area in the Non-Oral Malodor Group and Oral Malodor Group

Among the six areas of the tongue, there was greater tongue coating distribution in the oral malodor group than in the non-oral malodor group, especially in areas A, B, and C which are located posteriorly on the tongue. The middle posterior section of the tongue (B) had the highest tongue coating, followed by the lateral posterior areas (A and C). On the anterior of the tongue, areas D and E were similar and area F was the lowest. There were significant differences between both groups in all of the areas except area D (*p* < 0.05) (Table 2, Figure 4).

### 3.3. Comparison of Bacterial Activity, the Distribution of Tongue Coating, and VSCs in the Non-Oral Malodor Group and Oral Malodor Group

The oral malodor group showed significantly higher values than the non-oral malodor group with regards to bacterial activity and the distribution of tongue coating. Also, as a result of checking the level of halitosis, both H_2_S and CH_3_SH were higher in the oral malodor group than in the non-oral malodor group. However, there was a significant difference only in H_2_S (*p* < 0.05) (Table 3).

### 3.4. Associations between Oral Malodor, Bacterial Activity, and the Distribution of Tongue Coating

Table 4 shows the results confirming the correlation between bacterial activity, tongue coating distribution, and oral malodor in all participants. Bacterial activity and the distribution of tongue coating were determined to be highly correlated (r = 0.937, *p* < 0.01). Bacterial activity was associated with H_2_S (r = 0.240, *p* < 0.05), while there was no association between bacterial activity and CH_3_SH. The distribution of tongue coating was associated with H_2_S (r = 0.223, *p* < 0.01) but had no correlation with CH_3_SH. H_2_S and CH_3_SH were highly correlated (r = 0.796, *p* < 0.01).

### 3.5. Comparison of Tests for Bacterial Activity and the Distribution of Tongue Coating in the Diagnosis of Oral Malodor

The analysis for bacterial activity in the diagnosis of oral malodor showed the highest accuracy at AUC = 0.680, with cut-off >43 (Figure 5). The test for the distribution of tongue coating in the diagnosis of oral malodor showed the highest accuracy at AUC = 0.654, with cut-off >34. The bacterial activity test was more accurate than the test for the distribution of tongue coating, but there was no significant difference between the two tests.

## 4. Discussion

H_2_S and CH_3_SH are the main contributors to the intraoral causes of halitosis-derived VSCs [25]. In particular, H_2_S is associated with gastrointestinal disorders, and CH_3_SH is associated with periodontal diseases and tongue coating on the dorsal surface of the tongue [8,26,27]. Awano et al. [28] reported that CH_3_SH is the most important factor in oral malodor among VSCs and stated that it is important to control tongue coating, which is highly associated with CH_3_SH. Tonzetich et al. [14] reported that H_2_S and CH_3_SH were significantly reduced by removing the tongue coating on the surface of the posterior dorsal surface, which is difficult to manage. In addition, since it was reported that the greater the amount of tongue coating, the higher the concentration of intraoral VSCs [29], it is important to quantitatively assess tongue coating to manage oral malodor. Bosy et al. [18] proposed a method for classifying tongue coating as none, light, medium, and heavy. Gomez [30] divided the tongue into nine areas and scored the thickness of tongue coating by area, and Winkel [20] assessed tongue coating by dividing the tongue into six areas, scoring each area and summing up the scores. However, these methods may be limited by the subjective judgement of the examiners. Thus, this study aimed to validate the assessment of the distribution of tongue coating and bacterial activity in the diagnosis of oral malodor using the QLF technology-based Qraycam Pro and the TB01 1.05 Analyzer program.

The quantitative analysis of tongue coating using digital QLF images and by dividing the tongue into six areas using the TB01 1.05 Analyzer program showed that there was a significant difference in the anterior tongue only in two areas. On the other hand, the posterior tongue had a higher distribution of tongue coating in the oral malodor group than in the non-oral malodor group in all areas. Previous studies conducted by Tonzetich et al. and Jung et al. demonstrate that tongue coating, especially on posterior areas, is clearly correlated with oral malodor [13,31]. Thus, it is necessary to manage tongue coating on the posterior dorsal surface, which can be effectively performed with a tongue cleaner rather than a toothbrush [32]. Since the discontinuation of tongue coating control can result in a gradual increase in anaerobic bacteria in dental plaque, which produces more VSCs through metabolic interactions [33], it is important to consistently manage tongue coating to reduce bacterial growth.

In this study, bacterial activity and the distribution of tongue coating were higher in the oral malodor group than in the non-oral malodor group. In addition, oral malodor, bacterial activity, and distribution of tongue coating were related. Most significantly, the distribution of tongue coating and bacterial activity had a correlation with H_2_S but no correlation with CH_3_SH. This is in line with a previous study conducted by Han et al. [34] showing that the removal of tongue coating reduced CH_3_SH in people with healthy oral status but had no effect on CH_3_SH in patients with periodontal diseases. Since most of the participants in this study, whose mean age was 74 years, had periodontal diseases, CH_3_SH had no correlation with the distribution of tongue coating and bacterial activity. Nevertheless, this result is not consistent with a higher tendency of CH_3_SH in the group with more tongue coating than in the group with less tongue coating, as demonstrated in a previous study conducted by Ok et al. [35]. This warrants further investigation with an extended age range of participants.

Although tongue coating is known as the major cause of oral malodor, there is a marked lack of clinical trials of diagnostic tests using systematic, accurate, and quantitative analysis for its management. However, there are adequate standardized protocols for the management of tongue coating.

In this study, the images of the dorsal surface of the tongue were captured using Qraycam Pro, and the amounts of tongue coating and bacterial activity were quantitatively analyzed to determine their correlation with oral malodor. Therefore, follow-up observation using digital-QLF imagery is possible after tongue-coating care (tongue brushing and oral antisepsis) for halitosis patients in dental clinics. The results validated these tests for tongue-coating distribution and bacterial activity and indicate that a more systematic and accurate management of tongue coating and oral malodor can be achieved by actively using digital-QLF images to assess the distribution of tongue coating and bacterial activity.

This study is limited in that information on the participants’ systemic diseases, oral diseases, and medications was collected only through interviews; therefore, the results cannot be generalized. Thus, it is necessary to collect the participants’ data using a more accurate and objective method and to conduct an additional investigation that expands the age group. Nevertheless, this study is meaningful in that it confirms the possibility of using a digital QLF device in clinical practice for the diagnosis of tongue coating and bacterial activity, which is essential for the management of oral malodor.

## 5. Conclusions

Digital QLF images showed a higher distribution of tongue coating and bacterial activity in the oral malodor group than in the non-oral malodor group through quantitative analysis. In addition, since both tests showed significant accuracy in the assessment of oral malodor, systematic management of tongue coating and reduced oral malodor can be achieved by actively using digital QLF images and oral malodor measurements.

## Figures and Tables

**Figure 1 healthcare-11-00217-f001:**
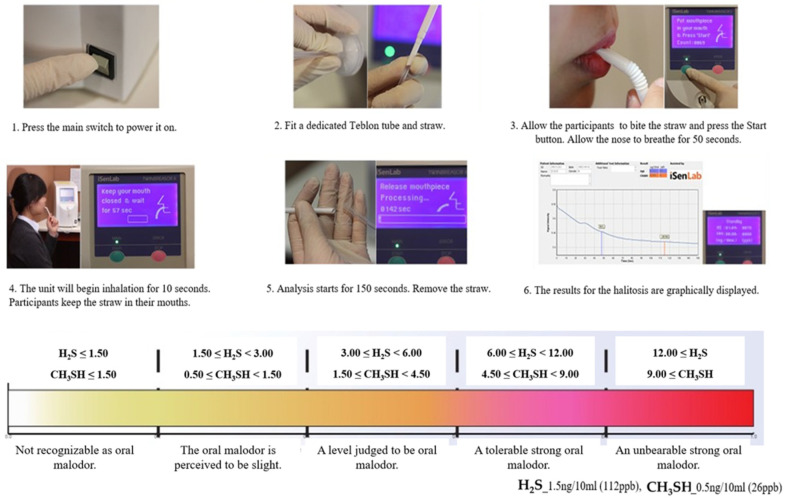
Oral malodor measurement method and evaluation method using Twin Breasor II.

**Figure 2 healthcare-11-00217-f002:**
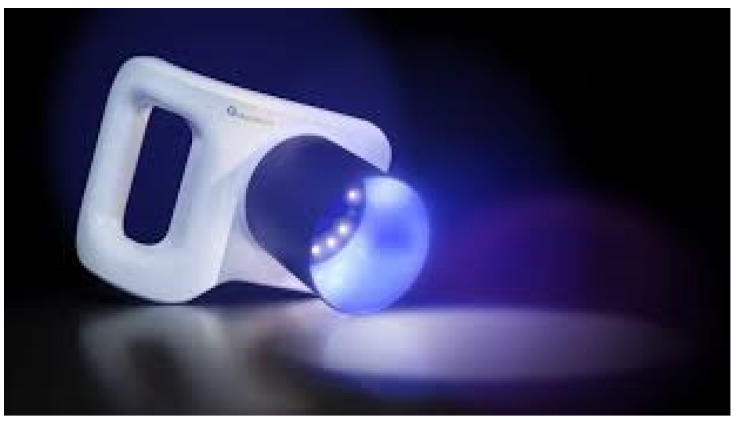
QLF device (Qraycam Pro, AIOBIO, Seoul, Republic of Korea).

**Figure 3 healthcare-11-00217-f003:**
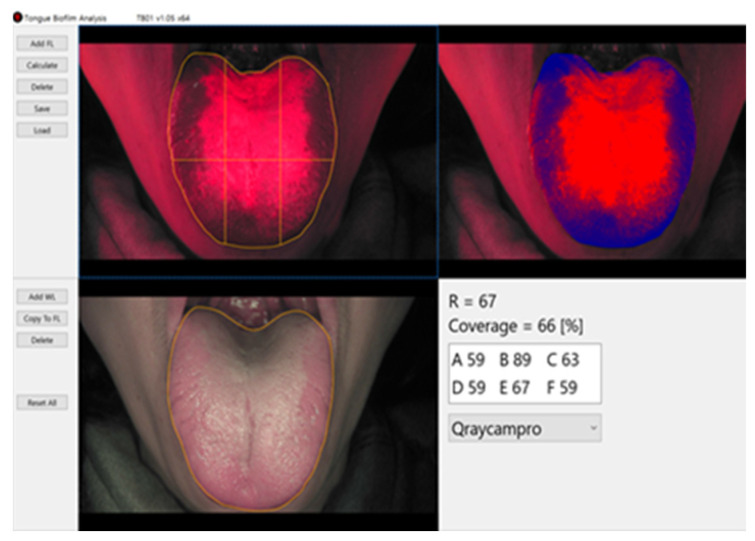
Evaluation of tongue coating distribution and bacterial activity using TB TB01 1.05 analysis program.

**Figure 4 healthcare-11-00217-f004:**
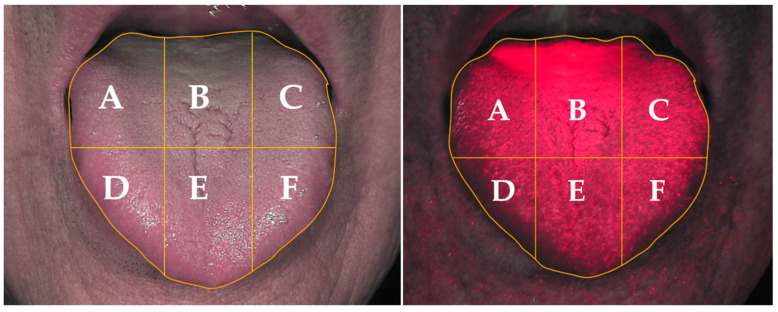
Tongue-coating distribution by tongue area following analysis.

**Figure 5 healthcare-11-00217-f005:**
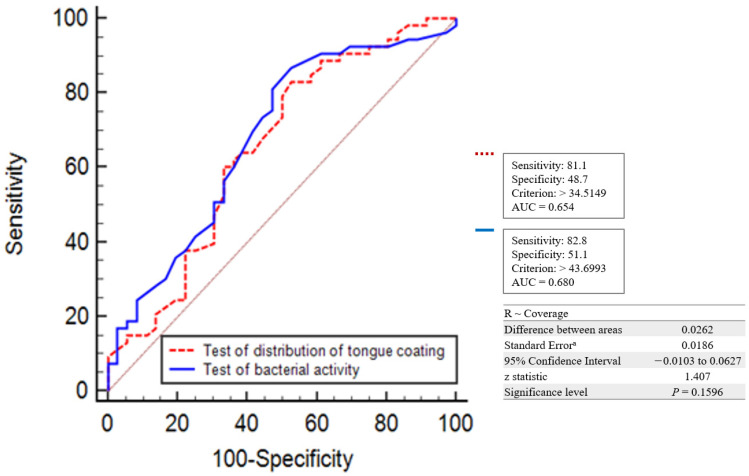
Comparison of accuracy between the test for the distribution of tongue coating and the test for bacterial activity.

**Table 1 healthcare-11-00217-t001:** Comparison of characteristics of the non-oral malodor group and the oral malodor group.

		Non-Oral Malodor Group (N = 36)	Oral Malodor Group (N = 53)	*p*
Gender	Male	15 (41.7)	21 (39.6)	1.000
	Female	21 (58.3)	32 (60.4)	
Age	≤65	21 (58.3)	31 (58.5)	1.000
	>65	15 (41.7)	22 (41.5)	
Type of cohabitation	Couple without children	13 (36.1)	22 (41.5)	0.839
	With children	8 (22.2)	12 (22.6)	
	Alone	15 (41.7)	19 (35.8)	
Taking medicine	No	13 (36.1)	17 (32.1)	0.890
	HBP ^a^	9 (25.0)	13 (24.5)	
	DM ^b^	4 (11.1)	9 (17.0)	
Drinking habits ^†^ (Mean ± SD)		1.39 ± 0.99	1.53 ± 1.09	0.540
Smoking habits ^†^ (Mean ± SD)		1.33 ± 0.79	1.26 ± 0.66	0.655
Exercise habits ^†^ (Mean ± SD)		2.58 ± 1.34	2.98 ± 1.19	0.144
Current health status ^†^ (Mean ± SD)		2.94 ± 1.15	2.89 ± 0.93	0.795

By chi-squared test: ^†^ independent *t*-test: *p* < 0.05. ^a^ Taking medication for high blood pressure. ^b^ Taking medication for diabetes. The participants with VSCs 0 were classified as the non-oral malodor group, and those with H_2_S ≥ 1.5 and CH_3_SH ≥ 0.5 were classified as the oral malodor group.

**Table 2 healthcare-11-00217-t002:** Comparison of distribution by tongue-coating area in non-oral malodor group and oral malodor group (Unit: %).

	Non-Oral Malodor Group (N = 36)	Oral Malodor Group (N = 53)	*p*
A	49.22 ± 13.12	55.47 ± 15.21	0.048 *
B	49.06 ± 15.78	60.77 ± 18.50	0.003 *
C	45.25 ± 13.34	51.89 ± 14.14	0.029 *
D	44.89 ± 10.24	49.42 ± 11.77	0.064
E	43.28 ± 9.75	49.49 ± 11.68	0.010 *
F	40.61 ± 9.89	45.53 ± 11.61	0.040 *

By independent *t*-test: * *p* < 0.05.

**Table 3 healthcare-11-00217-t003:** Comparison of bacterial activity, the distribution of tongue coating, and VSCs in the non-oral malodor group and oral malodor group.

	Non-Oral Malodor Group (N = 36)	Oral Malodor Group (N = 53)	*p*
Bacterial activity (%)	45.75 ± 10.24	52.70 ± 11.08	0.003 *
Distribution of tongue coating (%)	39.19 ± 23.15	51.66 ± 19.43	0.010 *
H_2_S (ppb)	0.00 ± 0.00	20.23 ± 41.17	0.004 *
CH_3_SH (ppb)	0.00 ± 0.00	7.40 ± 28.06	0.060

By independent *t*-test: * *p* < 0.05.

**Table 4 healthcare-11-00217-t004:** Associations of oral malodor, bacterial activity, and the distribution of tongue coating.

	Bacterial Activity	Distribution of Tongue Coating	H_2_S	CH_3_SH
Bacterial activity	1			
Distribution of tongue coating	0.937 **	1		
H_2_S	0.240 *	0.223 **	1	
CH_3_SH	0.141	0.150	0.796 **	1

By correlation analysis: ** *p* < 0.01, * *p* < 0.05.

## Data Availability

Not applicable.

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
