# Peer review of "Analysis of Quantitative Light-Induced Fluorescence Images for the Assessment of Bacterial Activity and Distribution of Tongue Coating"

_healthcare, 2023, doi:10.3390/healthcare11020217_

Round 1

Reviewer 1 Report

Please see uploaded pdf.

Author Response

Author's Reply is attached.

Thank you. 

Reviewer 2 Report

L11 : to verify and to validate the correlation…

L84-85 : idem

L111-112 : How the cumulate gases (H2S + CH3SH + (CH3)2S ) can have a 0 or 1 value ? May you  explain that better ?

L113-114 None of the information written on the graph or the screen can be read : are those pictures really useful ?

L124 : Can you differentiate the two parameters « quantitative evaluation of the tongue coating » and « bacterial activity » ? they seem to be both in relation with the red colour on Figure 3 ? Is the intensity of the red colour the traduction of the bacterial activity ? In that case, how have been evaluated the tongue coating ?

L152 : ≥ ?

L162 : « This is a table. Tables should be placed in the main text near to the first time they are cited » : Legend of Table 1 has to be confirmed

L185 : H2  ?

L211-212 : H2S is associated with tongue coating and CH3SH with periodontal disease (Yaegaki 1992)

This article is describing an interesting  way to diagnose the tongue coating and the bacterial activity through a sensor able to detect the red fluorescence produced by porphyrin metabolites secreted by intraoral bacteria. The results show a correlation between the high values measured by the sensor and the oral malodour group. This system may effectively be useful to limit the subjective evaluation of the practitioner when scoring the tongue coating just by a visual control.

Nevertheless, the microbial analysis of the tongue coatings in the two groups would have been an essential and complementary evaluation ; morever, it could have been an interesting way to explain the patients with good oral hygiene, no  periodontitis, few amount of tongue coating, and the presence of halitosis.

Author Response

Author's Reply is attached.

Thank you. 

Reviewer 3 Report

In the present research, the authors have tried to correlate the tongue coating and bacterial activity by using the Qraycam Pro technique. 

The manuscript is quite preliminary but well written and can be improved by adding following points.

Line no 91-92- A questionnaire retrieved after self administration.... Meaning of this line is not clear. Please rephrase it.

Line 95-96- Sample size was calculated to be 84 and 10 % droput was assumed. By this , the patient no to be included should be 92/93 Why 89 was included?? Any specific reason..

Line no 97-98- Approval number given by Institutional review board should also be suffixed with date of approval.

Section- 2.2 Crieteria to select the should be justified along with justifications. Questionnarie can also be included as supplementary file.

Section- 2.3- Is the method Used is novel?? If no, then provide reference.

Caption for figures should be elaborated to briefly describe the content of figure so that readers can have better understanding with quick overlook.

- In fig 2- What does the  HS and MM indicates. I cannot figure out this in text also.

Section 2.5- No reference for methodology adopted.

Line no 151-152 should be revised.

Results and discussion

Section 3.1- Results of Sociodemographic characteristics are not discussed at all.

In table 2: Significance of results can be indicated by * or any other indicator in table. 

Quantitative assessment of bacterial activity by Qraycom Pro images is done by analysis of red fluorescence. Is it a novel usage of software/instrument. If no, give the reference and also elaborate the method that how software quantify the data for bacterial activity (For example- any scoring pattern or how analysis of intensity of color fluorescence was done by software??

Section 3.4- Correlation coefficient for bacterial activity and H2S was shown to be 0.24 and author considered it to be significantly correlated. However for corelation, R value (Coefficient of corelation) should be near to 1 ( at least above 0.8). How they can justify, it is highly correlated??? 

Author Response

Author's Reply is attached.

Thank you.

Reviewer 4 Report

The topic of the present in silico study, evaluating qraycam pro images for the assessment of bacterial activity and distribution of tongue coating and its association with halitosis, is very interesting.

The issue has been widely described.

The manuscript is well-organized and written. Methods and Results are clearly presented. The discussion section is well structured, although it needs to be slightly expanded.

Reported findings currently presented may pave the way for further clinical investigations and may be clinically relevant.

Therefore, I would suggest adding considerations concerning:

-       coating tongue management (i.e., tongue brushing, oral antisepsis);

-       qraycam pro images use in follow-ups.

In addition, I would suggest better specify limits and strengths.

Have you excluded subjects suffering from periodontitis, active varies or mucosal infections at the time of the procedure; if so, please, state it in the text.

Author Response

Author's Reply is attached.

Thank you.

Round 2

Reviewer 1 Report

The revised manuscript is much improved, however there is still room for improvement. Please see reviewer comments and suggestions in authors revised pdf 

Author Response

We have attached the comment from the reviewer as a memo and explained it.

Thank you.

Reviewer 4 Report

I congratulate the Authors for the work done

Author Response

Thank you so much.